# Determinants of Public Knowledge, Attitude, and Practice on Antibiotic Use in Saudi Arabia: A Regional Cross-Sectional Study

**DOI:** 10.3390/healthcare13141666

**Published:** 2025-07-10

**Authors:** Wadia S. Alruqayb, Fahad H. Baali, Manar Althbiany, Alanoud Alharthi, Sara Alnefaie, Raghad Alhaji, Reem Alshehri, Wael Y. Khawagi, Monther A. Alshahrani, Hassan Arida, Abdullah A. Alshehri

**Affiliations:** 1Department of Clinical Pharmacy, College of Pharmacy, Taif University, Taif 21944, Saudi Arabiaw.khawagi@tu.edu.sa (W.Y.K.); 2College of Pharmacy, Taif University, Taif 21944, Saudi Arabia; 3Department of Pediatric Medicine, College of Medicine, Taif University, Taif 21944, Saudi Arabia; 4Department of Pharmaceutical Chemistry, College of Pharmacy, Taif University, Taif 21944, Saudi Arabia

**Keywords:** antibiotic use, public knowledge, health practices, antimicrobial resistance, western region, Saudi Arabia

## Abstract

**Background:** Antibiotic resistance (AMR) is a critical global and national health challenge, largely driven by the misuse and overuse of antibiotics. Understanding the public′s knowledge and practices regarding antibiotic use is essential for informing effective interventions. This study aimed to assess the levels of knowledge, attitude, and practice (KAP) related to antibiotic use among adults in Saudi Arabia’s Western Region and to identify the demographic and behavioral determinants of these outcomes. **Methods:** A regional cross-sectional survey was conducted from March to June 2025 using a 40-item self-administered online questionnaire. Adults aged ≥ 18 years residing in the Western Region of Saudi Arabia were recruited via social media using snowball sampling. Descriptive statistics and Chi-square tests were used to examine associations, while multivariate logistic regression was employed to identify determinants of high knowledge and good practices, presented as adjusted odds ratios (aOR) with 95% confidence intervals (CI). **Results:** A total of 891 participants were included; most were female (63.6%) and aged 18–30 years (56.2%). Moderate knowledge of antibiotic use was observed in 54.0% of participants, while 30.8% had high knowledge. In terms of attitude and practice, 55.6% demonstrated good performance and 42.8% average performance. High knowledge was significantly associated with the female gender (aOR = 1.90; 95% CI: 1.34–2.70), age of 41–50 years (aOR = 2.22; 95% CI: 1.42–3.48), and a postgraduate education (aOR = 15.37; 95% CI: 1.84–128.13). Good practices were associated with the female gender (aOR = 2.32; 95% CI: 1.66–3.24) and being married (aOR = 1.99; 95% CI: 1.43–2.77). A moderate positive correlation was found between knowledge and practice scores (r = 0.406, *p* < 0.001). **Conclusions:** Significant variability in public KAP regarding antibiotic use was identified. Female gender, older age, and higher education were key determinants of better KAP. These findings emphasize the need for targeted educational strategies focusing on high-risk groups to support rational antibiotic use and mitigate antimicrobial resistance.

## 1. Introduction

Antibiotics are among the most critical medical innovations of the 20th century, having transformed the treatment of bacterial infections and significantly reduced the associated morbidity and mortality [1,2]. The widespread use of antibiotics has enabled advances in surgery, cancer chemotherapy, and the management of infectious diseases [3,4]. However, the effectiveness of antibiotics is increasingly threatened by the global rise in antimicrobial resistance (AMR), with significant implications for public health and strains on healthcare systems [5,6]. AMR is largely driven by the inappropriate and excessive use of antibiotics including taking antibiotics without prescription, using them for viral infections, failing to complete prescribed courses, and sharing medications with others [7]. These behaviors are particularly prevalent in community settings and are often driven by misinformation, cultural practices, lack of awareness, or limited access to healthcare [8,9]. Research consistently supports the need for both regulatory measures and improved public education to combat this growing crisis [10].

Globally, numerous studies have highlighted the critical role of public knowledge, attitude, and practice (KAP) in influencing antibiotic use [11,12,13,14,15]. In many regions, including the Middle East, a large proportion of the population remains inadequately informed about the correct indications for antibiotics, the importance of completing prescribed courses, and the dangers associated with antibiotic misuse [12,16]. Misconceptions, such as the belief that antibiotics are effective against colds or influenza, remain prevalent despite ongoing awareness campaigns. These knowledge gaps often translate into suboptimal attitudes and risky behaviors that hinder efforts to combat AMR. In Saudi Arabia, the widespread availability of antibiotics, historically including over-the-counter access, has contributed to their frequent misuse [17]. Although recent policy changes have introduced stricter regulations around antibiotic dispensing, inappropriate antibiotic use remains a challenge [18].

Few studies have assessed KAP related to antibiotic use among the general population in Saudi Arabia [19,20]. Previous research has largely focused on individual cities, such as Jeddah, limiting the generalizability of findings to the wider Western Region [20]. Moreover, these studies lacked predictive analyses, which are crucial for identifying high-risk groups and informing targeted interventions. A comprehensive regional assessment is particularly important in the Western Region, which comprises both rural areas and major cities such as Makkah, Jeddah, and Madinah—key hubs for residents, pilgrims, tourists, and healthcare seekers from across the country and abroad. This demographic diversity presents distinct challenges and opportunities for effective public health planning. Therefore, this study aimed to assess KAP levels regarding antibiotic use among adults in the Western Region and to explore key sociodemographic predictors. By identifying factors associated with appropriate or inappropriate antibiotic use, the findings aim to inform targeted education efforts, public awareness initiatives, and antibiotic stewardship programs tailored to the region’s specific needs.

## 2. Methods

### 2.1. Study Design and Participants

This cross-sectional study was conducted over a three-month period (March to June 2025) using a self-administered online questionnaire. The target population included adults aged 18 years and older residing in the Western Region of Saudi Arabia who voluntarily agreed to participate. Eligibility criteria required participants to be able to understand either Arabic or English and to provide informed consent. Individuals were excluded if they were under 18 years of age or not residents of the Western Region.

### 2.2. Questionnaire Development

A structured, self-administered questionnaire was developed based on a review of the relevant literature and validated by expert review [12,19,20]. The questionnaire was created using Google Forms and included a preface outlining the study objectives, confidentiality details, researcher contact information, and a consent checkbox.

The distributed questionnaire consisted of 40 items divided into four sections. Section A included 9 closed-ended questions capturing sociodemographic characteristics. Section B assessed knowledge about antibiotics through 12 items with response options of “Yes,” “No,” or “I don’t know.” Correct answers were scored as 1 point, while incorrect or “I don’t know” responses were scored as 0. Total knowledge scores were categorized as follows: 0–4 (low knowledge), 5–8 (moderate knowledge), and 9–12 (high knowledge). Section C evaluated attitudes and practices related to antibiotic use using 13 items rated on a four-point Likert scale (Always, Sometimes, Rarely, and Never) to promote more decisive responses. Responses reflecting positive behaviors were scored as 1 and negative behaviors as 0. Total attitude/practice scores were classified as 0–4 (poor), 5–8 (average), and 9–12 (good). These cut-off points were based on the equal interval distribution of the total possible scores, a commonly applied method in KAP studies when standardized thresholds are not available [21]. This approach has been used in previous research evaluating public knowledge and practices related to antibiotic use and resistance. Section D comprised 6 items addressing sources of information and public awareness, presented in various formats including multiple choice, yes/no, and brief opinion-based responses. The estimated completion time for the questionnaire was 5–10 min. The original English version was translated into Arabic using a forward–backward translation method and reviewed by eight bilingual experts. The Arabic version showed strong face validity, with a scale content validity index (S-CVI) of 0.93 [22,23].

### 2.3. Piloting the Questionnaire

To assess the questionnaire’s clarity and practicality, a pilot test was carried out among 15 individuals from the target population, which is aligning with common practice for piloting self-administered surveys. This process aimed to evaluate the instrument’s readability, understanding, and overall feasibility. Minor revisions were made to the wording of some items based on participant feedback. Those who took part in the pilot study were excluded from the final data analysis.

Following data collection, internal consistency was assessed using Cronbach’s alpha, based on responses from the full study sample. The knowledge section (12 items) demonstrated acceptable reliability (α = 0.64), while the attitudes and practices section (13 items) showed good reliability (α = 0.70). These values indicate that the questionnaire items reliably measured the intended constructs.

### 2.4. Sample Size Calculation

The required sample size was calculated using the Raosoft online calculator, assuming a 95% confidence level, a 5% margin of error, and a 50% response distribution. Based on the estimated size of the population, which is approximately 10.5 million, according to the Saudi Government Authority for Statistics (GASTAT), and these parameters, the minimum sample size was estimated to be 385 participants.

### 2.5. Sample Recruitment and Data Collection

The online questionnaire was disseminated to targeted participants through popular social media platforms in Saudi Arabia, specifically WhatsApp and Telegram. Snowball sampling was employed to maximize reach, where participants were encouraged to share the questionnaire link with their network living in the Western Region.

### 2.6. Statistical Analysis

Descriptive statistics were used to summarize the data: categorical variables were presented as frequencies and percentages, and continuous variables as means and standard deviations. The normality of continuous variables was assessed using the Kolmogorov–Smirnov test. As most continuous variables were not normally distributed, the Kruskal–Wallis test was used for group comparisons. The Chi-square test was applied to assess associations between categorical variables. The Pearson correlation analysis was used to assess the relationship between total knowledge and attitude/practice scores, as these scores were approximately normally distributed.

Binary logistic regression was used to identify independent predictors of high knowledge and good attitude/practice levels. Both crude odds ratios (CORs) and adjusted odds ratios (aORs) with 95% confidence intervals (CIs) were reported. Variables with *p* < 0.20 in univariate logistic regression were included in the multivariate models. A *p*-value < 0.05 was considered statistically significant. Multicollinearity was assessed using the variance inflation factor (VIF), with values > 5 indicating high correlation. Data were analyzed using IBM SPSS Statistics version 25 (IBM Corp., Armonk, NY, USA).

### 2.7. Ethical Considerations

The study protocol was approved by the Ethics Committee of Taif University, Saudi Arabia (Application No. 46-092; Date: 27 November 2024). Informed consent was obtained electronically from each participant before completing the questionnaire. Participants were informed about the study objectives, and confidentiality and anonymity were ensured throughout the data collection and analysis process.

## 3. Results

### 3.1. Participants’ Demographic Characteristics

A total of 891 participants were included in the study. The majority were female (63.6%), and more than half (56.2%) were aged between 18 and 30 years. Most respondents held a bachelor’s degree (66.6%), followed by those with a high school education (23.0%). Over half of the participants were single (53.0%), and 41.1% were married. In terms of geographic distribution, the largest proportion of participants resided in Taif (53.6%), followed by Jeddah (18.9%). Monthly income varied, with 42.9% reporting an income of less than 3000 SAR. Regarding antibiotic use, 41.4% of participants reported using antibiotics 1–2 times in the past year, and 23.2% reported using them 3–5 times; 21.7% reported no antibiotic use during the same period. Government pharmacies were the primary source of antibiotics for 57.7% of participants. Additionally, 20.7% of respondents reported working in the healthcare sector. Table 1 summarizes the demographic characteristics of the participants.

### 3.2. Levels of Knowledge and Attitude/Practice Toward Antibiotic Use

Participants demonstrated varying levels of knowledge and attitude/practice regarding antibiotic use. Approximately 54.0% had moderate knowledge, while 30.8% had high knowledge, and 15.2% had low knowledge. In terms of attitude and practice, 55.6% of participants were classified as having good practices, 42.8% had average practices, and only 1.6% had poor practices. Table 2 and Figure 1 present the distribution of participants by levels of knowledge and attitude/practice toward antibiotic use.

### 3.3. Association Between Knowledge Level and Attitude/Practice Toward Antibiotic Use

Participants with high knowledge were much more likely to report good practices (79.3%), with only 20.4% showing average practices and 0.4% poor practices. In contrast, among those with low knowledge, only 34.1% demonstrated good practices, while the majority (63.0%) had average practices, and 3.0% exhibited poor practices.

A statistically significant association was found between categorized knowledge levels and attitude/practice levels (χ^2^, *p* < 0.001). In addition, the Pearson correlation analysis revealed a moderate positive relationship between total knowledge and attitude/practice scores (r = 0.406, *p* < 0.001) (Table 3).

### 3.4. Predictors of High Knowledge Toward Antibiotic Use

Binary logistic regression analysis was conducted to identify factors independently associated with having a good level of knowledge about antibiotic use and resistance (Table 4). Older age, female gender, and higher education were significantly associated with greater knowledge. Participants aged 31–40 (aOR = 1.67; 95% CI: 1.03–2.72), 41–50 (aOR = 2.22; 95% CI: 1.42–3.48), and >50 years (aOR = 2.13; 95% CI: 1.19–3.82) had higher odds of good knowledge compared to those aged 18–30. Females were more likely than males to be knowledgeable (aOR = 1.90; 95% CI: 1.34–2.70). Participants with higher education, particularly those with postgraduate degrees (aOR = 15.37; 95% CI: 1.84–128.13), also showed significantly better knowledge.

Frequent antibiotic use was negatively associated with knowledge, especially among those using antibiotics more than five times annually (aOR = 0.37; 95% CI: 0.21–0.65). Non-healthcare workers had lower odds of good knowledge (aOR = 0.14; 95% CI: 0.09–0.20). Surprisingly, those who did not receive doctor counseling were more likely to be knowledgeable (aOR = 1.78; 95% CI: 1.29–2.46). No significant associations were found for pharmacist counseling or sources of information.

### 3.5. Predictors of Good Attitude and Practice Toward Antibiotic Use

Binary logistic regression analysis was conducted to identify factors independently associated with having a good attitude toward antibiotic use (Table 5). Female gender (aOR = 2.32; 95% CI: 1.66–3.24), higher education (aOR = 2.78; 95% CI: 1.07–7.23 for high school), and being married (aOR = 1.99; 95% CI: 1.43–2.77) were significantly associated with better attitudes.

Participants residing in Madinah (aOR = 0.36; 95% CI: 0.16–0.82) or other regions (aOR = 0.37; 95% CI: 0.22–0.61) were less likely to have a good attitude compared to those in Taif. More frequent antibiotic use was negatively associated with attitude, especially among those using antibiotics more than five times annually (aOR = 0.26; 95% CI: 0.16–0.44).

Non-healthcare workers had lower odds of a good attitude (aOR = 0.58; 95% CI: 0.39–0.86). Participants relying on the internet (aOR = 0.46; 95% CI: 0.28–0.74) or media (aOR = 0.28; 95% CI: 0.09–0.89) as sources of information also showed poorer attitudes. Similarly to the knowledge model, those not receiving doctor counseling had better attitudes (aOR = 1.89; 95% CI: 1.35–2.56), while pharmacist counseling was not significantly associated.

## 4. Discussion

This study investigated the levels and determinants of knowledge and attitude/practice toward antibiotic use among the Western Saudi general population. The findings revealed substantial variability in knowledge and practices, with notable associations identified across demographic, behavioral, and informational factors. More than half of the participants demonstrated moderate knowledge, and nearly one-third had high knowledge regarding antibiotic use and resistance. Similarly, over half of the respondents exhibited good attitudes and practices. These findings suggest a relatively favorable level of public awareness, though notable gaps persist—particularly among specific subgroups.

The predominance of low to moderate knowledge levels aligns with findings from previous studies conducted in both Saudi and international contexts [19,20,24]. This highlights the ongoing need for public health education aimed at improving antibiotic-related knowledge, which in turn may promote appropriate use and reduce the risk of AMR.

Consistent with previous studies conducted in Saudi Arabia and other countries, our logistic regression analysis demonstrated that the female gender, older age, and a higher education level were significant predictors of better knowledge regarding antibiotic use [20,25,26,27]. Notably, participants aged over 40 years had more than twice the odds of demonstrating high knowledge compared to younger adults, a trend that may reflect cumulative life experience or more frequent interactions with healthcare providers. Education emerged as a particularly strong determinant, with those holding postgraduate degrees having more than 15 times the odds of possessing good knowledge relative to participants with less than a high school education—mirroring findings from studies in Ghana, South Korea, and Lebanon, which consistently show higher education levels are associated with greater antibiotic literacy [28,29,30].

Participants who reported frequent antibiotic use (more than five times annually) were significantly less likely to demonstrate good knowledge or practices regarding antibiotic use. This inverse relationship may reflect inappropriate or habitual antibiotic consumption, potentially driven by misinformation or lack of professional guidance, as observed in several community and student populations [7,31]. In contrast, individuals employed in the healthcare sector consistently demonstrated significantly higher knowledge and better practices, underscoring the role of professional exposure and training in fostering responsible antibiotic use [32,33].

Interestingly, participants who did not receive counseling from doctors were more likely to exhibit high knowledge and good practices regarding antibiotic use. Although this contrasts with conventional expectations, it may reflect a more proactive health information-seeking behavior among these individuals or indicate potential gaps in the quality of physician–patient communication [34,35]. For example, while many physicians feel confident in their counseling abilities, gaps remain in areas like infection prevention and the appropriate disposal of antibiotics [35,36].

In contrast, no significant associations were observed for pharmacist counseling, suggesting the need to re-evaluate the pharmacist’s role in patient education. Studies show that although pharmacist-led interventions can be highly effective in controlled settings [37], routine pharmacist counseling often lacks depth and coverage of essential information such as side effects or adherence [38]. This indicates a missed opportunity in community pharmacy settings to contribute meaningfully to antibiotic stewardship [39,40].

Participants’ residential region and sources of information also emerged as significant predictors of attitudes and practices. Individuals residing outside major urban centers, particularly in less populated areas, were found to have lower odds of demonstrating appropriate attitudes and practices toward antibiotic use. This likely reflects disparities in access to healthcare infrastructure, education, and outreach, consistent with findings from Ethiopia and the Philippines where rural populations exhibited lower knowledge and poorer practices [11,41]. Moreover, reliance on non-professional sources such as the internet or media was associated with poorer practices and misconceptions, emphasizing the critical need for targeted, credible public education campaigns [42,43].

### 4.1. Implications for Public Health and Policy

These findings underscore the urgent need for targeted educational interventions to improve antibiotic literacy, particularly among young adults, males, and individuals outside the healthcare sector. Prior research highlights that these groups often demonstrate poorer knowledge and practices related to antibiotic use [44,45]. Outreach should focus on correcting misconceptions among those who frequently use antibiotics and emphasizing the risks associated with misuse [42]. Moreover, enhancing the role of pharmacists in public health communication may improve the accessibility and credibility of antibiotic counseling, especially as pharmacist-led interventions have shown measurable improvements in knowledge and practices [46,47,48].

Interestingly, the inverse association observed with doctor counseling raises critical concerns about the quality and consistency of physician–patient communication. Although many doctors express confidence in their counseling abilities, studies suggest that time constraints and communication gaps may reduce effectiveness [49,50]. Further qualitative research is needed to explore whether patient expectations, provider workload, or systemic barriers influence these outcomes. These patterns, along with previous research among healthcare students, point to the value of reinforcing antibiotic-related education early in professional training to ensure future providers are better equipped to counsel patients effectively [21].

### 4.2. Strengths and Limitations

A key strength of this study is the use of both univariate and multivariate logistic regression models, which allowed for the robust identification of independent predictors. Additionally, the relatively large and diverse sample enhances the generalizability of the findings within the Western Region of Saudi Arabia. However, several limitations must be acknowledged. First, reliance on self-reported data introduces potential for recall bias and social desirability bias, which may affect the accuracy of responses [51,52]. Second, although the survey included a broad range of demographic variables, unmeasured factors such as health literacy, previous infection history, or access to healthcare information may have influenced participants’ knowledge and behavior but were not accounted for in the analysis [53]. Third, the questionnaire did not capture whether antibiotics obtained from government or private pharmacies were accessed via prescription, under pharmacist supervision, or over the counter. This limits interpretation of the regulatory context and access-related influences on antibiotic use. Future research should address this aspect to better understand its implications for responsible antibiotic use.

## 5. Conclusions

This study highlights critical sociodemographic and behavioral determinants of antibiotic-related knowledge and practices in the Western Saudi population. Efforts to promote appropriate antibiotic use should focus on tailored health education, especially for high-risk groups. Strengthening the role of community healthcare providers and improving communication strategies are essential steps toward mitigating antibiotic misuse and addressing antimicrobial resistance. Further national research is needed to capture geographic and demographic variations across all regions of Saudi Arabia, enabling healthcare authorities to design equitable, region-specific interventions.

## Figures and Tables

**Figure 1 healthcare-13-01666-f001:**
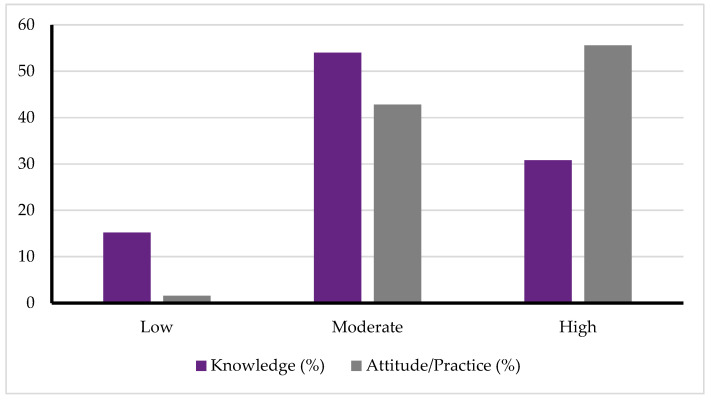
Distribution of participants’ knowledge and attitude/practice levels regarding antibiotic use.

**Table 1 healthcare-13-01666-t001:** Demographic characteristics of study participants (*n* = 891).

Characteristics		*n* (%)
Gender		
	Male	324 (36.4)
	Female	567 (63.6)
Age (years)		
	18–30	501 (56.2)
	31–40	142 (15.9)
	41–50	170 (19.1)
	>50	78 (8.8)
Education Level		
	Less than high school	26 (2.9)
	High school	205 (23.0)
	Bachelor’s degree	593 (66.6)
	Postgraduate studies	67 (7.5)
Marital Status		
	Single	472 (53.0)
	Married	366 (41.1)
	Divorced	37 (4.1)
	Widow	16 (1.8)
Residency City/Area		
	Taif	478 (53.6)
	Makkah	64 (7.2)
	Jeddah	168 (18.9)
	Yanbu	33 (3.7)
	Madinah	45 (5.1)
	Other	103 (11.6)
Monthly Income (SAR)		
	Less than 3000	382 (42.9)
	3000 to 10,000	255 (28.6)
	More than 10,000	254 (28.5)
Antibiotic Use in the Past Year		
	1–2 times	369 (41.4)
	3–5 times	207 (23.2)
	More than 5 times	122 (13.7)
	None	193 (21.7)
Source of Antibiotics		
	Government pharmacies	514 (57.7)
	Private pharmacies	377 (42.3)
Employment in Healthcare Sector		
	Yes	184 (20.7)
	No	707 (79.3)

**Table 2 healthcare-13-01666-t002:** Distribution of knowledge and attitude/practice levels toward antibiotic use among participants.

Domain	Level	(*n*)	(%)	Mean ± SD
Knowledge	Low (<5)	135	15.2	7.11 ± 2.59
	Moderate (5–8)	481	54.0	
	High (>8)	275	30.8	
Attitude/Practice	Poor (<5)	14	1.6	8.70 ± 2.19
	Average (5–8)	381	42.8	
	Good (>8)	496	55.6	

**Table 3 healthcare-13-01666-t003:** Distribution of attitude/practice levels by knowledge category among participants.

Knowledge Level	*n* (%)	Poor Attitude/Practice	Average Attitude/Practice	Good Attitude/Practice	*p*-Value
Low (<5)	135 (15.2%)	4 (3.0%)	85 (63.0%)	46 (34.1%)	<0.001
Moderate (5–8)	481 (54.0%)	9 (1.9%)	240 (49.9%)	232 (48.2%)	
High (>8)	275 (30.8%)	1 (0.4%)	56 (20.4%)	218 (79.3%)	

**Table 4 healthcare-13-01666-t004:** Logistic regression analysis of independent factors against knowledge of participants.

	Risk factors	Good Level of Knowledge (%)	COR (95% CI)	*p* Value	aOR (95% CI)	*p* Value
Age						
	18–30	159 (31.7%)	1		1	
	31–40	37 (26.1%)	0.76 (0.50–1.15)	0.195	1.67 (1.03–2.72)	0.038
	41–50	52 (30.6%)	0.95 (0.65–1.38)	0.781	2.22 (1.42–3.48)	0.001
	More than 50	27 (34.6%)	1.14 (0.69–1.88)	0.613	2.13 (1.19–3.82)	0.011
Gender						
	Male	78 (24.1%)	1			
	Female	197 (34.7%)	1.68 (1.23–2.28)	0.001	1.90 (1.34–2.70)	<0.001
Education Level						
	Less than high school	1 (3.8%)	1		1	
	High school	40 (19.5%)	6.06 (0.80–46.07)	0.082	9.50 (1.18–76.44)	0.034
	Bachelor’s degree	210 (35.4%)	13.71 (1.84–101.88)	0.011	14.48 (1.85–113.41)	0.011
	Postgraduate studies	24 (35.8%)	13.95 (1.78–109.50)	0.012	15.37 (1.84–128.13)	0.012
Marital Status						
	Single	149 (31.6%)	1			
	Married	114 (31.1%)	0.98 (0.73–1.32)	0.897		
	Divorced	8 (21.6%)	0.60 (0.27–1.34)	0.211		
	Widow	4 (25.0%)	0.72 (0.23–2.28)	0.579		
Residence						
	Taif	170 (35.6%)	1			
	Makkah	17 (26.6%)	0.66 (0.37–1.18)	0.157		
	Jeddah	49 (29.2%)	0.75 (0.51–1.09)	0.133		
	Yanbu	8 (24.2%)	0.58 (0.26–1.31)	0.191		
	Madinah	8 (17.8%)	0.39 (0.18–0.86)	0.020		
	Other	23 (22.3%)	0.52 (0.32–0.86)	0.011		
Monthly Income						
	Less than 3000 SAR	105 (27.5%)	1			
	3000 to 10,000 SAR	80 (31.4%)	1.21 (0.85–1.71)	0.290		
	More than 10,000 SAR	90 (35.4%)	1.45 (1.03–2.04)	0.034		
Antibiotic Use in the Past Year						
	1–2 times	139 (37.7%)	1		1	
	3–5 times	47 (22.7%)	0.49 (0.33–0.72)	<0.001	0.59 (0.38–0.89)	0.013
	More than 5 times	21 (17.2%)	0.34 (0.21–0.58)	<0.001	0.37 (0.21–0.65)	0.001
	None	68 (35.2%)	0.90 (0.63–1.29)	0.570	0.96 (0.64–1.45)	0.856
Working in Healthcare						
	Yes	116 (63.0%)	1		1	
	No	159 (22.5%)	0.17 (0.12–0.24)	<0.001	0.14 (0.09–0.20)	<0.001
Source of Information						
	The doctor	135 (29.2%)	1			
	The pharmacist	88 (32.7%)	1.18 (0.85–1.63)	0.314		
	Internet	44 (37.6%)	1.46 (0.96–2.24)	0.078		
	Family or friends	5 (21.7%)	0.68 (0.25–1.86)	0.446		
	Media (TV, newspapers)	3 (15.8%)	0.46 (0.13–1.59)	0.217		
Doctor Counseling						
	Yes	118 (24.3%)	1		1	
	No	157 (38.8%)	1.97 (1.48–2.63)	<0.001	1.78 (1.29–2.46)	<0.001
Pharmacist Counseling						
	Yes	191 (30.2%)	1			
	No	84 (32.4%)	1.11 (0.81–1.52)	0.517		

**Table 5 healthcare-13-01666-t005:** Logistic regression analysis of independent factors against practices of participants.

	Risk Factors	Good Level of Knowledge (%)	COR (95% CI)	*p* Value	AOR (95% CI)	*p* Value
Age						
	18–30	228 (45.5%)	1		1	
	31–40	66 (46.5%)	1.04 (0.72–1.51)	0.838		
	41–50	85 (50.0%)	1.20 (0.85–1.70)	0.311		
	More than 50	41 (52.6%)	1.33 (0.82–2.14)	0.246		
Gender						
	Male	111 (34.3%)	1		1	
	Female	309 (54.5%)	2.30 (1.73–3.05)	<0.001	2.32 (1.66–3.24)	<0.001
Education level						
	Less than high school	8 (30.8%)	1		1	
	High school	93 (45.4%)	0.80 (0.30–2.10)	0.646	2.78 (1.07–7.23)	0.036
	Bachelor’s degree	295 (49.7%)	1.49 (0.84–2.63)	0.172	2.51 (0.99–6.31)	0.051
	Postgraduate studies	24 (35.8%)	1.77 (1.05–3.00)	0.032	1.63 (0.55–4.71)	0.380
Marital status						
	Single	213 (45.1%)	1		1	
	Married	193 (52.71%)	1.36 (1.03–1.78)	0.029	1.99 (1.43–2.77)	<0.001
	Divorced	8 (21.6%)	0.34 (0.15–0.75)	0.008	0.51 (0.21–1.27)	0.147
	Widow	6 (37.5%)	0.73 (0.26–2.04)	0.548	1.08 (0.33–3.47)	0.903
Residence						
	Taif	258 (54.0%)	1		1	
	Makkah	28 (43.8%)	0.66 (0.39–1.12)	0.126	0.75 (0.42–1.33)	0.324
	Jeddah	86 (51.2%)	0.89 (0.63–1.27)	0.534	0.97 (0.65–1.44)	0.867
	Yanbu	10 (30.3%)	0.37 (0.17–0.80)	0.011	0.72 (0.30–1.74)	0.467
	Madinah	9 (20.0%)	0.21 (0.10–0.45)	<0.001	0.36 (0.16–0.82)	0.015
	Other	29 (28.2%)	0.33 (0.21–0.53)	<0.001	0.37 (0.22–0.61)	<0.001
Monthly income						
	Less than 3000 SAR	173 (45.3%)	1			
	3000 to 10,000 SAR	120 (47.1%)	1.07 (0.78–1.48)	0.660		
	More than 10,000 SAR	127 (50.0%)	1.21 (0.88–1.66)	0.244		
Frequency of antibiotic use						
	1–2 times	203 (55.0%)	1		1	
	3–5 times	77 (37.2%)	0.48 (0.34–0.69)	<0.001	0.52 (0.36–0.77)	0.001
	More than 5 times	28 (23.0%)	0.24 (0.15–0.39)	<0.001	0.26 (0.16–0.44)	<0.001
	None	112 (58.0%)	1.13 (0.80–1.61)	0.494	1.35 (0.92–1.99)	0.125
Working in healthcare						
	Yes	104 (56.5%)	1		1	
	No	316 (44.7%)	0.62 (0.45–0.86)	0.004	0.58 (0.39–0.86)	0.006
Source of information						
	The doctor	239 (51.6%)	1		1	
	The pharmacist	125 (46.5%)	0.81 (0.60–1.10)	0.179	0.77 (0.53–1.11)	0.155
	Internet	46 (39.3%)	0.61 (0.40–0.92)	0.018	0.46 (0.28–0.74)	0.001
	Family or friends	6 (26.1%)	0.33 (0.13–0.85)	0.022	0.58 (0.21–1.62)	0.298
	Media (TV, newspapers)	4 (21.1%)	0.25 (0.08–0.76)	0.015	0.28 (0.09–0.89)	0.033
Doctor counseling						
	Yes	202 (41.6%)	1		1	
	No	218 (53.8%)	1.64 (1.26–2.14)	<0.001	1.89 (1.35–2.56)	<0.001
Pharmacist counseling						
	Yes	302 (47.8%)	1		1	
	No	118 (45.6%)	0.91 (0.68–1.22)	0.546	0.72 (0.50–1.04)	0.078

## Data Availability

The datasets used and analyzed during the current study are available from the corresponding author on reasonable request.

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
