# Peer review of "Determinants of Public Knowledge, Attitude, and Practice on Antibiotic Use in Saudi Arabia: A Regional Cross-Sectional Study"

_healthcare, 2025, doi:10.3390/healthcare13141666_

Round 1
Reviewer 1 Report
Comments and Suggestions for Authors
The study offers valuable regional insights into antibiotic-related knowledge and behavior in Saudi Arabia. While KAP studies on this topic are common, this study adds value through multivariate analysis and a large regional sample. Highlighting how this work advances prior literature would further justify its contribution. The following additional clarifications are needed to strengthen the manuscript:
The authors state that participants were recruited via social media using snowball sampling, but they do not specify which platforms were used or how the survey was distributed. Clarifying this would help assess potential sampling biases and the representativeness of the sample.
Please clarify whether antibiotics from government and private pharmacies were obtained via prescription, pharmacist supervision, or over the counter. This is important for interpreting access and regulation. Access without supervision could relate to poorer practices or misconceptions.
The finding that those without doctor counseling had better KAP is interesting. Briefly describing their characteristics (e.g., age, education, employment sector) would help clarify why this association exists.
Author Response
We sincerely appreciate your time and effort in reviewing our manuscript. Please find our detailed responses below, along with the corresponding revisions, which are highlighted and tracked in the resubmitted files.
The study offers valuable regional insights into antibiotic-related knowledge and behavior in Saudi Arabia. While KAP studies on this topic are common, this study adds value through multivariate analysis and a large regional sample. Highlighting how this work advances prior literature would further justify its contribution. The following additional clarifications are needed to strengthen the manuscript:
- The authors state that participants were recruited via social media using snowball sampling, but they do not specify which platforms were used or how the survey was distributed. Clarifying this would help assess potential sampling biases and the representativeness of the sample.
Thank you for this insightful comment. We agree that providing more detail on the recruitment strategy enhances the clarity and transparency of the study. We have revised the Methods section to specify that the questionnaire was disseminated via popular social media platforms in Saudi Arabia, specifically WhatsApp and Telegram.
“The online questionnaire was disseminated to targeted participants through popular social media platforms in Saudi Arabia, specifically WhatsApp and Telegram.”
- Please clarify whether antibiotics from government and private pharmacies were obtained via prescription, pharmacist supervision, or over the counter. This is important for interpreting access and regulation. Access without supervision could relate to poorer practices or misconceptions.
Thank you for this important observation. We acknowledge the significance of clarifying the context of antibiotic access in interpreting the regulatory and behavioral implications. However, our questionnaire only asked participants to report the source of antibiotics (i.e., government or private pharmacies), without capturing whether they were obtained by prescription, under pharmacist supervision, or over the counter. We recognize this as a limitation that may hinder interpretation of access and regulatory oversight. Accordingly, we have revised the Strengths and Limitations section of the manuscript to explicitly acknowledge this limitation and suggest that future studies explore this issue in more detail to better assess its impact on antibiotic use behaviors.
“Third, the questionnaire did not capture whether antibiotics obtained from government or private pharmacies were accessed via prescription, under pharmacist supervision, or over the counter. This limits interpretation of the regulatory context and access-related in-fluences on antibiotic use. Future research should address this aspect to better understand its implications for responsible antibiotic use.”
- The finding that those without doctor counseling had better KAP is interesting. Briefly describing their characteristics (e.g., age, education, employment sector) would help clarify why this association exists.
Thank you for this thoughtful observation. We agree that further analysis of participant characteristics could offer additional insights into this unexpected finding. However, our primary analysis focused on broader trends and associations, and subgroup comparisons were beyond the intended scope of this paper. We acknowledge the value of such exploratory analyses and will consider this direction in future work to better understand the factors influencing KAP in the absence of direct counseling.
Reviewer 2 Report
Comments and Suggestions for Authors
This study is captivating, especially considering the increasing levels of AMR and the overuse of antibiotics; having the right knowledge, attitudes, and practices is essential for optimizing antibiotic use. I have some suggestions for revision in the manuscript, which are as follows
- Kindly justify or provide the reason for the small sample size of the pilot study.
- Provide details on evaluating the internal consistency or reliability of the questionnaire.
- Provide a reason for using a 4-point Likert scale instead of a 5-point Likert scale.
- Provide a reference or justification for categorizing knowledge scores: 0–4 (low knowledge), 5–8 (moderate knowledge), and 9–12 (high knowledge and for categorizing attitude/practice scores as 0–4 (poor), 5–8 (average), and 9–12 (good).
- Include a figure in your manuscript to highlight some key findings, if possible.
Author Response
We sincerely appreciate your time and effort in reviewing our manuscript. Please find our detailed responses below, along with the corresponding revisions, which are highlighted and tracked in the resubmitted files.
This study is captivating, especially considering the increasing levels of AMR and the overuse of antibiotics; having the right knowledge, attitudes, and practices is essential for optimizing antibiotic use. I have some suggestions for revision in the manuscript, which are as follows
1. Kindly justify or provide the reason for the small sample size of the pilot study.
Thank you for your comment. The pilot study included 15 participants, which aligns with commonly accepted practices for piloting self-administered questionnaires. The primary purpose of the pilot was to evaluate the clarity, relevance, and comprehensibility of the survey items, as well as the layout and functionality of the online format. A sample size of 10–30 participants is typically considered sufficient for identifying issues with wording, structure, and user experience in such contexts. We have added this justification to the Methods section of the manuscript and clarified that pilot participants were excluded from the main analysis.
“To assess the questionnaire’s clarity and practicality, a pilot test was carried out among 15 individuals from the target population, which is aligning with common practice for piloting self-administer surveys. This process aimed to evaluate the instrument’s readability, understanding, and overall feasibility. Minor revisions were made to the wording of some items based on participant feedback. Those who took part in the pilot study were excluded from the final data analysis.”
2. Provide details on evaluating the internal consistency or reliability of the questionnaire.
Thank you for this valuable comment. We assessed the internal consistency of the questionnaire using Cronbach’s alpha. The results showed acceptable reliability for the knowledge section (α = 0.64) and good reliability for the attitudes and practices section (α = 0.70). These values indicate that the items within each section consistently measure the intended constructs. We have now included this information in the revised manuscript under the Methods section (subsection: “Survey Instrument and Pilot Testing”).
“Following data collection, internal consistency was assessed using Cronbach’s alpha, based on responses from the full study sample. The knowledge section (12 items) demonstrated acceptable reliability (α = 0.64), while the attitudes and practices section (13 items) showed good reliability (α = 0.70). These values indicate that the questionnaire items reliably measured the intended constructs.”
3. Provide a reason for using a 4-point Likert scale instead of a 5-point Likert scale.
Thank you for raising this important point. We intentionally used a 4-point Likert scale to evaluate attitudes and practices to avoid neutral responses and encourage participants to express a more definitive position. This forced-choice format is commonly used in behavioral research to reduce central tendency bias and enhance the interpretability of responses. We have added this rationale to the Questionnaire Development subsection in the Methods section.
4. Provide a reference or justification for categorizing knowledge scores: 0–4 (low knowledge), 5–8 (moderate knowledge), and 9–12 (high knowledge and for categorizing attitude/practice scores as 0–4 (poor), 5–8 (average), and 9–12 (good).
Thank you for this helpful comment. The categorization of knowledge, attitude, and practice (KAP) scores into low/moderate/high or poor/average/good was based on an equal interval distribution of the total possible score ranges. This is a commonly used approach in KAP studies when standardized or validated cut-off points are not available. We have now clarified this rationale in the Methods section and added supporting references.
These cut-off points were based on equal interval distribution of the total possible scores, a commonly applied method in KAP studies when standardized thresholds are not available [21]. This approach has been used in previous research evaluating public knowledge and practices related to antibiotic use and resistance.
5. Include a figure in your manuscript to highlight some key findings, if possible.
Thank you for the valuable suggestion. In response, we have included a new figure in the revised manuscript that visually presents the distribution of participants’ knowledge and attitude/practice levels regarding antibiotic use. This figure complements the data presented in Table 2 and helps to highlight key findings in a more accessible format. The figure has been added to the Results section and appropriately cited in the text.
Reviewer 3 Report
Comments and Suggestions for Authors
Thank you for giving me the opportunity to review this cross-sectional study entitled:
"This study aimed to assess the levels of knowledge, attitude and practice (KAP) related to antibiotic use among adults in Saudi Arabia’s Western Region and to identify demographic and behavioural determinants of these outcomes."
The manuscript is generally well written and addresses an important topic. However, I believe several revisions are necessary to improve the clarity and methodological rigour:
Methodology:
Please provide a clear explanation of the criteria used to assign scores for good knowledge, poor knowledge, etc. Indicate the basis for these thresholds and reference any relevant guidelines or literature.
Sample Size Calculation:
The description of the sample size estimation is incomplete. Kindly clarify on which numeric value or estimated proportion of the adult population in the Western Region the calculation was based.
Table 5:
The formatting of Table 5 requires correction, as it currently appears unclear and difficult to interpret. Please revise to improve readability.
References:
Ensure that all references are formatted according to the journal’s author guidelines.
Author Response
We sincerely appreciate your time and effort in reviewing our manuscript. Please find our detailed responses below, along with the corresponding revisions, which are highlighted and tracked in the resubmitted files.
Thank you for giving me the opportunity to review this cross-sectional study entitled:
"This study aimed to assess the levels of knowledge, attitude and practice (KAP) related to antibiotic use among adults in Saudi Arabia’s Western Region and to identify demographic and behavioural determinants of these outcomes."
The manuscript is generally well written and addresses an important topic. However, I believe several revisions are necessary to improve the clarity and methodological rigour:
- Methodology:
1.1 Please provide a clear explanation of the criteria used to assign scores for good knowledge, poor knowledge, etc. Indicate the basis for these thresholds and reference any relevant guidelines or literature.
Thank you for your comment. The categorization of knowledge, attitude, and practice (KAP) scores into levels such as low/moderate/high or poor/average/good was based on equal interval distribution of the total possible score ranges. This approach is commonly used in KAP studies when standardized thresholds are not available, as it facilitates interpretation and group comparisons. We have clarified this rationale in the Methods section and added a supporting reference.
These cut-off points were based on equal interval distribution of the total possible scores, a commonly applied method in KAP studies when standardized thresholds are not available [21]. This approach has been used in previous research evaluating public knowledge and practices related to antibiotic use and resistance.
1.2 Sample Size Calculation:
1.3 The description of the sample size estimation is incomplete. Kindly clarify on which numeric value or estimated proportion of the adult population in the Western Region the calculation was based.
Thank you for your insightful comment. The sample size was calculated using the Raosoft online calculator, based on a 95% confidence level, 5% margin of error, and 50% response distribution. The population estimate used in the calculation was approximately 10.5 million adults residing in the Western Region of Saudi Arabia, as reported by the General Authority for Statistics (GASTAT)—the official agency responsible for statistical data in the Kingdom. We have revised the Methods section to clarify this point.
“The required sample size was calculated using the Raosoft online calculator, as-suming a 95% confidence level, a 5% margin of error, and a 50% response distribution. Based on the estimated size of population, which is approximately 10.5 million, according to the Saudi Government Authority for Statistics (GASTAT) and these parameters, the minimum sample size was estimated to be 385 participants.”
1.4 Table 5: The formatting of Table 5 requires correction, as it currently appears unclear and difficult to interpret. Please revise to improve readability.
Thank you for highlighting this important issue. We have carefully revised the formatting of Table 5 to enhance its readability and ensure that the data are clearly presented. Adjustments included aligning column headers, clarifying variable labels, and improving the layout to distinguish between groups and outcome categories. The updated table has been included in the revised manuscript.
- References:
2.1 Ensure that all references are formatted according to the journal’s author guidelines.
Thank you for your valuable advice. All references were carefully reviewed, ensuring formatting to be fully compliant with the journal guidelines.